# The Assessment of Housing Conditions, Management, Animal-Based Measure of Dairy Goats’ Welfare and Its Association with Productive and Reproductive Traits

**DOI:** 10.3390/ani9110893

**Published:** 2019-11-01

**Authors:** Francesco Tiezzi, Laura Tomassone, Gilberto Mancin, Paolo Cornale, Martina Tarantola

**Affiliations:** 1Veterinary Practitioner, 10100 Torino, Italy; f.tiezzi@libero.it (F.T.); gilberto.mancin@gmail.com (G.M.); 2Department of Veterinary Science, University of Turin, L.go Braccini, 2, 10095 Grugliasco (TO), Italy; laura.tomassone@unito.it; 3Department of Agricultural, Forest and Food Sciences, University of Turin, L.go Braccini, 2, 10095 Grugliasco (TO), Italy; paolo.cornale@unito.it

**Keywords:** dairy goats, farming systems, welfare assessment, productive and reproductive traits

## Abstract

**Simple Summary:**

Small ruminant production systems are generally perceived by consumers to be associated with a high standard of welfare, notably in relation to aspects of traditional breeding. However, their performances have reached the same levels of the dairy cow sector, with similar negative externalities. We aimed to evaluate the welfare of dairy goats of 32 farms located in Northwestern Italy by the application of an on-farm assessment protocol. The farms were classified as ‘intensive’ or ‘semi-intensive’ according to the access to pasture. Overall, we observed an adequate level of animal wellbeing both in intensive and semi-intensive farming systems. This is possible thanks to the increased knowledge on goat breeding characteristics, and to the fact that veterinarians and farmers understood the importance of welfare protection to achieve a better health, although some challenges remain, such as the farmers’ approach toward animals. The importance of the farmer’s role is highlighted by the positive association between the milk yield and the presence of the owner on the farm.

**Abstract:**

The aim of the study was to evaluate the welfare of dairy goats of 32 farms located in Northwestern Italy, applying an on-farm assessment protocol, centered on animal-, resources- and management-based measures. The farms were classified as ‘intensive’ or ‘semi-intensive’ according to access to pasture. During each on-farm visit, a checklist was compiled, based on specific scores for housing and management conditions, and animals’ nutritional status, health, and behavior. Finally, the possible association between welfare measures and productive and reproductive traits was assessed. Overall, we observed an adequate level of animal wellbeing both in intensive and semi-intensive farming systems. This is possible thanks to the increased knowledge on goat breeding characteristics, and to the fact that veterinarians and farmers understood the importance of welfare protection to achieve a better health. Higher milk production was associated to some management practices (presence of the owner on the farm, high frequency of bedding change), and to seasonal breeding (which was mainly performed in the intensive farming). Moreover, it was associated to a quantity of urea in the milk comprised between 33 and 44 mg/dL. In intensive farms, the prevalence of caseous lymphadenitis was significantly higher compared to non-intensive farms. The semi-intensive breeding system positively influences the animals’ behavior.

## 1. Introduction

Goat intensive production systems have spread through the northern countries of the Mediterranean basin and specialized dairy flocks have increased in size. Currently, nearly 1 billion goats are bred around the world; the milk and meat of goats can be used in functional foods and constitute very good basic products for developing the functional values of foods for people [1]. The evolution in productivity of sheep and goats and their role in agricultural development and human nutrition has been the focus of comprehensive studies in recent years, in order to provide stimuli and support for further improvement [2,3].

Small ruminant production systems are generally perceived by consumers to be associated with a high standard of welfare, notably in relation to aspects of traditional breeding. However, their performances have reached the same levels of the dairy cow sector, with similar negative externalities [4]. The negative impact of intensification of farming systems can be observed at several levels: development of specific infectious diseases (e.g., Caprine Arthritis Encephalitis Virus (CAEV)) or metabolic diseases, lower microbiological quality of milk [5], aggressive behavior, increased environmental pollution, and poor animal health [6]. Furthermore, goats and sheep are often managed by shepherds who have no specific skills and professional competences to be aware of the welfare standards of the animals [7].

Welfare is a multidimensional concept that requires all the component dimensions to be checked by specific indicators. It embraces absence of suffering, high levels of biological functioning—including absence of diseases—and the potential for animals to have ‘positive experience’ [8,9].

Welfare assessment systems include two broad categories: animal-based measures (e.g., body condition, disease state, level of injury, and flight distance) and resource-based measures (e.g., stockperson, environmental and genetic attributes, enclosure design features, access to litter) [10,11]. Both measurement categories are important. Recently, efforts have been devoted to the development and validation of systems for the evaluation of welfare through animal-based measures, such as the European Animal Welfare Indicators (AWIN), created for sheep, goat, horses, donkeys, and turkeys [12]. Indeed, significant advances in the welfare of small ruminants have recently been achieved [13,14,15].

In the present study, an on-farm welfare monitoring protocol was used to evaluate housing conditions, management and animal-based measures of dairy goat breeding and to assess if welfare measures were associated with productive and reproductive traits.

## 2. Materials and Methods

A total of 32 dairy goat farms were selected in the Piedmont and Liguria regions, Northwestern Italy. Convenience sampling was performed, based on the availability of the farms in the study area and willingness of farmers to collaborate. The farms were classified as an ‘intensive’ or ‘semi-intensive’ farming system according to the access to pasture. In semi-intensive farms, animals pastured for around three hours per day, from March to October.

The farms were visited once, in the period September 2014–May 2015, by two trained veterinarians who examined together each farm in about one hour and half.

In order to assess the welfare of the dairy goats, a specific on-farm protocol (Appendix A
Table A1, Table A2, Table A3) was created using parameters identified by experts and relevant in the published literature for meeting the animals’ needs and ensuring their well-being, nutritional status, good health, and good behavior. The checklist had specific scores for animal-based measures, resource-based measures and management-based measures.

Animals were subjected to clinical examinations in order to identify the health status.

On each farm, data on reproductive parameters were collected, in particular regarding the out of season breeding and synchronization of estrus, herd replacements, conception rating, kidding rating, deliveries per goat, and the percentage of does which were not pregnant 3 months after the introduction of the buck (at ecographic examination). Moreover, additional data were collected on milk quality traits, e.g., production per lactation period (in kilograms), lactation length (in days), and milk composition: percentage of fat, protein, lactose, casein; somatic cell count; colony forming units (cfu) per ml; urea concentration (mg/dL).

Data were analyzed by R software (Available online: https://www.r-project.org/; accessed on 20 October 2018) [16]. Parametric (*t*-test) and non-parametric (Kruskal–Wallis, Wilcoxon Sum Rank test) tests were used to evaluate the differences among continuous parameters and scores and assess their association with reproductive parameters. The chi-square test was used to study the association among categorical variables. The results were considered statistically significant when *p* < 0.05.

## 3. Results

### 3.1. Housing and Environment

The number of animals per farm ranged from 26 to 1024, with 18–620 lactating goats. Goats belonged to the breeds Saanen, ‘Roccaverano’, ‘Camosciata delle Alpi’, ‘Valdostana’, and mixed breeds. Only seven farms had more than 500 animals, and two bred pure Saanen or ‘Camosciata delle Alpi’.

According to the absence or presence of the pasture, 21 farms were classified as ‘intensive farming’ and 11 as ‘semi-intensive farming’. In semi-intensive farming, animals shared the pasture with wildlife in 81.8% of the cases, and the space allowance was 11 animals per hectare. The three hours spent on the pasture allowed goats to exercise after and before the feeding.

With regard to the number of lactating goats and goat kids, in the intensive farms, the average was 194 lactating goats (67 animals per pen) and 54 goat kids, and in the semi-intensive farms 83 (49 animals per pen) and 30, respectively. The average space allowance per animal was 2 m^2^. Overall, 90% of the farms had a separate area for the goat kids, with a space/kid ratio of <0.4 m^2^ in 62% of intensive farms and 75% of semi-intensive farms.

The illumination of the structures was considered adequate in 31 out of 32 farms, while ventilation was not adequate in 6 out of 21 for the intensive farms and 1 out of 11 in semi-intensive farms. Considering dustiness, it was adequate in 6 out of 21 in intensive farms and in 4 out of 11 in semi-intensive farms.

Non-slip floors were present on all semi-intensive farms and only in 52% of intensive farms.

Straw bedding was present on all farms. The bedding was changed with a frequency up to 4 times per month, with a mean of 1.5 times per month in the intensive farms and 0.8 in the semi-intensive. The frequency at which the bedding was changed was associated to milk production: In the 11 farms in which the frequency was higher than once per month, the average milk production per goat per year was significantly higher (*t*-test, *p* = 0.04), with 800 kg, versus 660 kg in the remaining 21 farms. This corresponds to a 21.2% increased production. A significant difference was also found in the mean milk production per goat per day (2.7 L/day versus 2.3 L; *t*-test, *p* = 0.04).

In general, animals could easily access the feeding troughs (on 96.9% of farms) and drinking places (90.6%). Drinking places were enough in number in all intensive farms and in 75% of the semi-intensive ones.

The cleaning of the troughs and of the farm in general (walls and floors) was given a medium score, with few farms with dirty or very clean structures (Table 1).

### 3.2. Management

As regards the management of the farms, the owner was not the manager in just 3 cases (2 intensive farms, 1 semi-intensive). A significantly higher annual milk yield was observed when the owner was present and managed the farm (720 vs. 500 kg/year; *t*-test *p* = 0.03).

The number of workers employed on the selected farms was low, with a maximum of 3–4 full-time employees in four farms with more than 400 goats. On 10 intensive farms, only one stockman managed the farm. In 8 out of 11 semi-intensive systems, at least 2 persons were present. Only 55% of the intensive farms had personnel with a specific education; this percentage grew to 75% in the semi-intensive farms.

The approach of the farmer towards the animals had lower scores in intensive farms compared with semi-intensive; the lowest score was attributed to 42.8% of the intensive farms versus 9.1% of the semi-intensive (Table 1).

Overall, 90.5% of intensive farms fed the animals with complete pelleted rations (18% crude protein, 9% fiber), versus 82% of the semi-intensive farms.

The other intensive farms used ad libitum “total mixed ration” containing 40% alfalfa hay and 60% concentrate consisting of ground barley, corn, salt, soya bean meal (CP 18%, 14% fiber).

The remaining semi-intensive farms used a mash mixture composed of soybean meal (44%), sunflower meal (28%), corn, barley, and beet pulp (18% crude protein, 10% fiber).

Correct cleaning procedures at the milking were adopted on 86% of the intensive farms and 83% semi-intensive farms, while the cleanliness of the udder was always good. However, goats showing signs of mastitis were found in 52% of intensive farms and 63.6% of semi-intensive farms. Other frequently detected diseases were respiratory illnesses and CAEV (52.4% and 45.4% of farms, respectively). The prevalence of caseous lymphadenitis was significantly higher in intensive farms (76.1% of intensive farms, 27.3% of semi-intensive farms; chi-squared test *p* < 0.01).

Antiparasitic treatments were administered in all intensive farms and 90.6% of the farmers vaccinated against clostridiosis.

Animal longevity was good in all farms, with goats over 6 years of age in 65.0% of intensive farms and in 81.8% of semi-intensive farms.

Mortality was lower than 5% in all intensive farms and in 90.9% of semi-intensive farms.

### 3.3. Animal-Based Measures

A description of animal-based variables is summarized in Table 2.

No significant differences were observed in these parameters between the two farming systems, even though most intensive farms had a medium-high score as regards the approach of the animals towards the farmers, while 36.4% of semi-intensive farms got the lower score.

Milk quality traits are described in Table 3.

A quantity of urea in the milk comprising between 33 and 44 mg/dL corresponded to a significantly higher milk production (820 vs. 610 kg/year; *t*-test *p* = 0.02).

### 3.4. Reproductive Parameters

As regards reproductive parameters, 53.1% of the farms (n = 17) performed out-of-season breeding (71.4% of intensive and 18.2% of semi-intensive farms), and 6.2% (n = 2) a synchronization of ovulation. Out-of-season breeding was associated to a significantly higher average milk production per goat per year (770 vs. 640 kg; *t*-test, *p* = 0.04), but with a lower number of deliveries per goat (5.0 vs. 6.0; Wilcoxon Sum Rank test, *p* = 0.02). On the other side, a higher mean number of deliveries per goat was observed in the semi-intensive rearing system (6.4) versus the intensive ones (5.0; Wilcoxon Sum Rank test, *p* < 0.01).

Artificial insemination was carried out in 15.6% of farms (14.3% of intensive and 18.2% of semi-intensive).

On all farms, the herd replacement was in-house, and the bucks were used to detect the female heats. Selected reproductive parameters are shown in Table 4.

## 4. Discussion

Our research aimed to increase the knowledge on dairy goats’ welfare considering different management systems, in a study area (Northwestern, Italy) where the interest on goat breeding is growing.

Our results show that both farming systems provide acceptable levels of welfare to the animals: Animal longevity was scored as good in all farms, and mortality was lower than 5%. However, some housing structural weaknesses were present in all farms, such as a poor air quality, which could explain the presence of signs of mastitis in almost 50% of farms, as suggested by Bergonier et al. [17], although the cleanness of milking procedures and of the udder was good.

The size of the examined semi-intensive farms, in which animals have access to the pasture, was smaller than that of the intensive farms. This implies a different farming management. Fewer breeders ran the intensive system, and the relationship between the stockperson and the animals was closer than in the semi-intensive farms, where the animals spend time outside. This was confirmed by the data on the latency period to the first contact between goat and assessor: Better scores were attributed to the intensive system, where the contact rate between humans and animals is higher, and goats are less suspicious and more used to meeting different people. Our results are in agreement with those of Can et al. [18], who had a better result in the tests carried out in large farms. On the other side, the farmer approach toward animals was better in the semi-intensive farms. The importance of the farmer’s role was confirmed by the positive association between milk yield and the presence of the owner in the farm, in accordance with literature [19].

The somatic cells counts (SCC) showed a high variability within the farms and do not indicate the presence of mastitis [20]. The total microbial count (TMC) in the intensive farms was higher compared to that in the semi-intensive, but the milk yield in full lactation was not different between the two systems. However, the relationship between TMC and SCC has yet to be clarified [21].

The frequency of change of the animals bedding also seems to influence the milk production, since environmental conditions can affect the udder health [22]. In the selected farms, milk yield was not influenced by other factors such as goat breed (almost all farms had mixed breeds), or parity and nutrition (which did not differ among farms).

A relationship between milk urea concentration and milk yield was detected, as observed by Hojman et al. [23] in cattle. Further studies could help to understand the ideal level of urea associated to an increased milk production, and how to maintain such levels. Despite the importance of milk urea as a nutritional indicator in dairy cows and sheep, very few studies have evaluated its use in dairy goats. In fact, degradability of dietary crude protein (CP) seems to be less important in goats than in dairy cattle, probably because of the higher rumen passage rate, which reduces the proportion of degraded CP and urea production [24].

Animal-based parameters like BCS, hair coat conditions, integument cleanness, and severe lameness provide information on the care of farmers for animals. We did not observe any difference in these health indicators between the two breeding systems, which generally received good scores. Antiparasitic treatments were administered in all the farms, and this could explain the good BCS [25]. BCS was not influenced by the presence of caseous lymphadenitis, which affected animals in several farms; this is in accordance with the results of Muri et al. [26]. The higher prevalence of this disease in the intensive farms can be due to the closer contacts among animals, since the disease is transmitted through contamination of superficial skin, wounds, or possibly by airborne transmission.

The space allowance per animal was 2 m^2^ per adult animal, which is the same as that recommended by Sevi et al. [27]. However, the number of animals and the absence of pasture influenced the presence of aggressive behavior, which was more prevalent in the intensive farms compared to the semi-intensive, where the presence of pasture allows more freedom of movements and the expression of normal social behavior [28]. The number of feed and water places was adequate; only in a few farms did we observe goats queuing. A sufficient number of places helps to reduce aggressive interactions [29].

Vocalizations are indicators of social isolation [28]. We observed a higher number of goats vocalizing in the intensive farms, where there were also more oblivious goats.

The animals did not show any stereotypies.

Reproductive seasonality is a limiting factor for the market, since both meat and milk industries are subjected to growing demands for constant production [30]. As regards reproductive parameters, out of season breeding was mainly performed in the intensive farms, and it was shown to be associated to a significantly higher average milk production. However, this was at the expense of life expectancy, which was shorter in intensive farms.

## 5. Conclusions

The study included goat farms that differed in their management. Our results confirm that the semi-intensive farming is still widespread; this system better enables the expression of normal social activities and positively affects animals’ behavior, thanks to the possibility for animals to have access to the pasture.

However, in both intensive and semi-intensive farming systems, we found an adequate level of animal wellbeing. Such good welfare is possible thanks to the increased knowledge on goat farming characteristics, and to the fact that veterinarians and farmers understood the importance of welfare protection in order to achieve a better health.

We underline the importance of evaluating some critical points of dairy goat welfare, such as the farmers’ approach toward animals and the cleanness of the bedding, which have a beneficial impact on milk production.

## Figures and Tables

**Table 1 animals-09-00893-t001:** Selected resources and management-based measures scored in intensive and semi-intensive goat farms (see Appendix A
Table A1 and Table A2), Northwestern Italy.

Scoring	Intensive Farms (n = 21)	Semi-Intensive Farms (n = 11)
Cleanliness of the floors
Pt. 1: >75%	3	0
Pt. 2: 25–75%	15	11
Pt. 3: <25%	3	0
Cleanliness of the feeding troughs
Pt.1: >75%	6	3
Pt.2: 25–75%	12	8
Pt.3: <25%	3	0
Cleanliness of the walls
Pt.1: >75%	5	1
Pt.2: 25–75%	10	7
Pt.3: <25%	6	3
Cleanliness of the bedding material in the lying area
Pt.1: very dirty	2	0
Pt.2: dirty	4	4
Pt.3: clean	9	3
NA	6	4
The farmers’ approach behavior
Pt.1: strongly negative physical interactions, with or without negative verbal interactions/avoids immediately, difficult to catch.	9	1
Pt.2: mild negative physical interaction/mild fear: attempts to avoid stockperson, but no pain	4	4
Pt.3: positive physical and verbal interaction/positive reaction, no fear; approaching stockperson immediately and initiating physical contact	8	6

Pt. = points assigned; NA: data not available.

**Table 2 animals-09-00893-t002:** Animal-based measures scored in intensive and semi-intensive goat farms, Northwestern Italy.

Score	Intensive Farms (n = 21)	Semi-Intensive Farms (n = 11)
Body condition score BCS		
Pt 1: Very thin	0	0
Pt 2: Regular	16	11
Pt 3: Very fat	5	0
Integument cleanliness		
Pt 1: Optimal	0	0
Pt 2: Medium	12	6
Pt 3: Poor	9	5
Hair coat condition, lesions		
No	14	10
Yes	7	1
Udder and teat hygiene		
Pt 1: Clean	21	11
Pt 2: Dirty	0	0
Pt 3: Very dirty	0	0
Severe lameness		
No	17	10
Yes	4	1
Oblivion		
Pt 1: 1–2% of animals	10	3
Pt 2: 3–5% of animals	2	1
Pt 3: >5% of animals	9	7
Relax		
Pt 1: >30% of animals	17	11
Pt 2: <30% of animals	4	0
Aggressiveness		
Pt 1: <5% of animals	9	7
Pt 2: 5–30%of animals	4	1
Pt 3: >30% of animals	8	3
Animals competition (queuing)		
No	17	10
Yes	4	1
Vocalizations		
No	16	10
Yes	5	1
Stereotypies		
No	19	11
Yes	2	0
Latency to the first contact		
Pt 1: >300 s	1	4
Pt 2: 120–300 s	15	6
Pt 3: < 120 s	5	1

Pt. = points assigned.

**Table 3 animals-09-00893-t003:** Milk yield and milk quality traits (mean ± SD) recorded in intensive and semi-intensive goat farms, Northwestern Italy.

Milk Trait	Intensive (n = 21)	Semi-Intensive (n = 11)
Milk yield in full lactation (kg)	740 ± 180	640 ± 160
Milking period (d)	290.5 ± 22.2	274.5 ± 23.4
Quality of milk:		
Fat (%)	3.8 ± 0.8	3.6 ± 0.6
Protein (%)	3.4 ± 0.3	3.4 ± 0.4
Lactose (%)	4.4 ± 0.2	4.5 ± 0.2
Casein (%)	2.7 ± 0.2	2.6 ± 0.7
Somatic cell content (cells/mL)	1,493,308	1,134,000
Colony forming units (cfu/mL)	115,166 ± 84,100	37,250 ± 17,500
Urea (mg/dL)	43.6 ± 11.5	36.7 ±9.8

**Table 4 animals-09-00893-t004:** Reproductive parameters registered in intensive and semi-intensive goat farms, Northwestern Italy.

Percentage Range	Intensive Farms (n = 21)	Semi-Intensive Farms (n = 11)
% of does not pregnant/year		
0–3%	4	2
3–5%	8	4
5–10%	6	4
10–30%	2	1
30–50%	1	0
Kidding rate		
Mean ± SD	98%	98%
Conception rating		
Mean ± SD	0.95 ± 0.03	0.97 ± 0.02
Deliveries/goat *		
Mean ± SD	5.0 ± 0.7	6.4 ± 1.3
Days of lactation		
Mean ± SD	290.5 ± 22.2	274.5 ± 23.4

* Significant difference.

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
