# Peer review of "The Assessment of Housing Conditions, Management, Animal-Based Measure of Dairy Goats’ Welfare and Its Association with Productive and Reproductive Traits"

_animals, 2019, doi:10.3390/ani9110893_

Round 1
Reviewer 1 Report
The topic is important not only for goat breeders from Italy but worldwide, discussing a plethora of influential factors on health, welfare and production in dairy goats which lately gained more and more terrain on the food market. The authors need to better organise the presentation of their results, be more explicit in English, this way improving the scientific impact of their research. In materials and methods, numerous statistical tests are mentioned, with no reference to the obtained results by applying each of them. The presentation of the measures in Appendix A is somewhat unclear (ie, Farmers number, farmers and educational levels, farmers experience. yes/no - what does that mean?).
English needs editing for a better understanding of the text by readers.

Author Response
We thank the reviewer for his extensive and thoughtful comments.
We believe we have addressed all of the major and minor comments that were raised by the reviewers and, in doing so, have crafted a paper that is more rigorous in content and clearer in presentation.
First, we copy below the reviewer’s major concerns and then we describe how we have addressed each of the issues.
Please see the attachment for the minor comments.
The authors need to better organise the presentation of their results, be more explicit in English, this way improving the scientific impact of their research.Answer. As indicated by the reviewer, we have rewritten the results. All results presented in the revised version of the manuscript, as well as in tables, have been updated.
In materials and methods, numerous statistical tests are mentioned, with no reference to the obtained results by applying each of them.Answer. Thanks. We have mentioned the statistical tests used in the ‘results’ section, together with their p-values.
The presentation of the measures in Appendix A is somewhat unclear (ie, Farmers number, farmers and educational levels, farmers experience. yes/no - what does that mean?).Answer. Thanks. We have added more information in Appendix A
English needs editing for a better understanding of the text by readers.Answer. Thank you, we have revised the English language

Reviewer 2 Report
Did you analyse the feed rations?
Please present the feed rations structure.
Did you do some blood analysis?
How did you establish the caseous lymphadenitis diagnosis?
Author Response
We thank the reviewer for his extensive and thoughtful comments.
We believe we have addressed all of the major and minor comments that were raised by the reviewers and, in doing so, have crafted a paper that is more rigorous in content and clearer in presentation.
First, we copy below the reviewer’s major concerns and then we describe how we have addressed each of the issues.
Did you analyse the feed rations? Please present the feed rations structure.
Answer: Thank you for this suggestion. We didn’t analyze the feed rations, but data about main constituents (e.g. CP, Fiber) were collected during the farms visit (we added rations structure in line 148)
Did you do some blood analysis?Answer: No. We did not
How did you establish the caseous lymphadenitis diagnosis?Answer: Vets performed clinical examinations on farms to identify clinical signs of diseases; we added a sentence in line 88
Reviewer 3 Report
This is an interesting paper, but experimental design does not allow to evaluate the differences in milk yield, in relation to welfare parameters. This is due to the presence of 5 goat breeds with genetic merit more different, moreover milk yield mainly depends to feeding. For this reason all comments that regard milk yield variation should be omitted.
Author Response
We thank the reviewer for his extensive and thoughtful comments.
We believe we have addressed all of the major and minor comments that were raised by the reviewers and, in doing so, have crafted a paper that is more rigorous in content and clearer in presentation.
First, we copy below the reviewer’s major concerns, and then we describe how we have addressed each of the issues.
This is an interesting paper, but experimental design does not allow to evaluate the differences in milk yield, in relation to welfare parameters. This is due to the presence of 5 goat breeds with genetic merit more different, moreover milk yield mainly depends to feeding. For this reason all comments that regard milk yield variation should be omitted.
Answer: The authors agree with the Reviewer comment that the considered goat breeds have different genetic merit. However, the present study did not aim at assessing differences in milk yield among breeds. Furthermore, almost all of the farms reared different breeds and mixed breeds, milk yields significantly differed among farms breeding the same breeds. This presence of multiple breeds in the same farm, together with the great high variability in goat milk yield, led milk yield not being influenced by breed (lines 228).
Concerning the effect of feeding on milk yield, we strongly agree with the Reviewer comment. However, as we detailed in a previous answer, feeding did not differ among farms, as shown by data regarding main constituents (lines 148).
For the above-mentioned reasons, we would propose to maintain the comments about milk yield
Round 2
Reviewer 1 Report
Thank the authors for considering and operating the suggested changes, thus increasing the clearness and scientific soundness of the paper